# Racial differences in testing for infectious diseases: An analysis of jail intake data

**Alysse G. Wurcel**[1,2], **Rubeen Guardado**[1], **Emily D. Grussing**[1,2]*, **Peter J. Koutoujian**[3], **Kashif Siddiqi**[3], **Thomas Senst**[3], **Sabrina A. Assoumou**[4,5], **Karen M. Freund**[2,6], **Curt G. Beckwith**[7]

**1** Department of Medicine Tufts Medical Center, Division of Geographic Medicine and Infectious Diseases, Boston, MA, United States of America, **2** Tufts University School of Medicine, Boston, MA, United States of America, **3** Middlesex Sheriff's Office, Medford, MA, United States of America, **4** Boston University School of Medicine, Boston, MA, United States of America, **5** Boston Medical Center, Boston, MA, United States of America, **6** Department of Medicine, Tufts Medical Center, Boston, MA, United States of America, **7** The Miriam Hospital/Alpert Medical School of Brown University, Providence, Rhode Island, United States of America

* Emily.grussing@tufts.edu

**Data Availability Statement:** Data cannot be shared publicly because of restrictions with the data being potentially identifiable and the sensitive nature of the data for people who are incarcerated.

## Abstract

HIV and hepatitis C virus (HCV) testing for all people in jail is recommended by the CDC. In the community, there are barriers to HIV and HCV testing for minoritized people. We examined the relationship between race and infectious diseases (HIV, HCV, syphilis) testing in one Massachusetts jail, Middlesex House of Corrections (MHOC). This is a retrospective analysis of people incarcerated at MHOC who opted-in to infectious diseases testing between 2016–2020. Variables of interest were race/ethnicity, self-identified history of psychiatric illness, and ever having experienced restrictive housing. Twenty-three percent (1,688/8,467) of people who were incarcerated requested testing at intake. Of those, only 38% received testing. Black non-Hispanic (25%) and Hispanic people (30%) were more likely to request testing than white people (19%). Hispanic people (16%, AOR 1.69(1.24–2.29)) were more likely to receive a test result compared to their white non-Hispanic (8%, AOR 1.54(1.10–2.15)) counterparts. Black non-Hispanic and Hispanic people were more likely to opt-in to and complete infectious disease testing than white people. These findings could be related to racial disparities in access to care in the community. Additionally, just over one-third of people who requested testing received it, underscoring that there is room for improvement in ensuring testing is completed. We hope our collaborative efforts with jail professionals can encourage other cross-disciplinary investigations.

## Introduction

Increasing hepatitis C virus (HCV) and HIV testing in jails is necessary to move towards the ultimate goal of global HCV and HIV eradication [1, 2]. Testing is the first step in the continuum of care, followed by linkage to treatment and either cure (HCV) or viral suppression (HIV) [3–8]. HIV and HCV testing in jails is not only cost-effective [9], but it is the standard of care recommended by the Centers for Diseases Control and Prevention [10–12].

Data are available from Tufts Medical Center for researchers who meet the criteria for access to confidential data. Please contact Dr. Alysse Wurcel (alysse.wurcel@tuftsmedicine.org) to access the data.

**Funding:** AGW:K08HS026008-01A, National Institute of Health. https://www.ahrq.gov/funding/training-grants/k-awards.html. The funders had no role in study design, data collection and analysis, decision to publish, or preparation of the manuscript.

**Competing interests:** The authors have declared that no competing interests exist.

Unfortunately, though, a recent nation-wide survey found that only 7% of jails offer HIV testing and only 5% offer HCV testing at intake [13]. Testing for HIV and HCV in jails, which typically detain persons for $\leq$ 2.5 years, can identify people with these illnesses and link people to treatment, preventing illness and prolonging life [14].

Examination of best practices for HIV and HCV testing in jails has revealed several lessons guiding the timing of offering testing (intake vs. exit testing), who gets testing (risk-based vs. universal) [13], the type of test (rapid vs. standard), the language used to offer testing (opt-out vs. opt in) [10.14], and the utility of coupling tests like HIV and HCV together [14–18]. Increasingly, there is a movement to deliberately examine the equity of testing interventions in order to overcome barriers that may prevent people who are Black or Hispanic from accessing HIV and HCV testing and treatment [19, 20]. Minoritized populations, including Black and Hispanic people, make up an increasing proportion of new HCV and HIV diagnoses [21–25].

Disparities in access to HIV and HCV testing have been reported in the community for people who are Black, Hispanic, Asian, and Indigenous [26–29]. Specifically, Black men who have sex with men and transgender women with history of incarceration face barriers to accessing HIV testing [30]. In a randomized controlled trial of rapid HIV testing versus standard testing, Black patients at a detoxification treatment center were less likely to receive their HIV results [31]. Intensifying already existing barriers to testing, the COVID19 pandemic disrupted harm reduction services [32] and led to increased HIV and HCV transmission in outbreaks reported across the U.S. [33–35].

We sought to purposefully examine the role of race/ethnicity in access to HIV/HCV testing at a jail. Massachusetts Department of Public Health (MassDPH) offers free infectious diseases testing to jails in MA including Middlesex County Jail, including Middlesex House of Correction (MHOC) in North Billerica, MA. We hypothesized that the Black non-Hispanic and Hispanic people who were incarcerated at MHOC would be less likely to opt-in to infectious diseases testing and, if testing was requested, less likely to have testing completed.

## Methods

The current study is a retrospective analysis of people incarcerated and detained at MHOC from January 1, 2016 to December 31, 2020. The MHOC is located less than 45 minutes away from the cities of Lowell and Lawrence, two 2015–2018 epicenters of HIV outbreaks in people who inject drugs [36]. The MassDPH funds a community-based public health agency to come into MHOC weekly and perform testing. HIV antibody testing was offered until 2019, when the DPH began to offer combined testing for HIV, HCV, and syphilis. At MHOC, infectious diseases testing is offered as "opt-in" method at intake with the question, "Do you want to be tested for HIV, hepatitis C, and syphilis?" to all persons entering the facility. If the person said yes, they are placed on a list to meet with the employee of the DPH-contracted group who visits the jails about two times a week. The employee obtains consent for HIV, HCV, and syphilis and draw a single tube of blood for testing. The blood specimen is sent to the MassDPH for testing, and the results are sent back to the jail. The MHOC nurse enters the data directly into the electronic medical record (EMR) in a form called the "Infectious Diseases (ID) Results" form and notifies the person about their results. Outside of the jail intake nursing evaluation, there are other opportunities for people to request testing, including at the time of the intake physical exam (offered within one month of arriving at the jail, but often refused by people who are incarcerated), or by writing a note ("sick slip") to the medical providers to request a test. The DPH also funds linkage-to-care specialists who help link people with HIV and HCV to care in the community after release. These individuals come to the jail 1–2 times a week and also manage linkage to HCV and HIV care in the community.

The database for this study was created by combining information from two separate sources: the Offender Management System (OMS) and the electronic medical record (EMR) used at the MHOC. The OMS is overseen by the Sheriff's Office and had demographic and administrative details about the people who were incarcerated during the study period. The information is self-reported and gathered by the individual booking the person into the carceral system. Variables drawn from the OMS included date of birth, dates of incarceration, dates of release, country of birth, and race/ethnicity (grouped) as entered in the database with options of white non-Hispanic, Black non-Hispanic, Hispanic, Asian, Pacific Islander, American Indian, Other, and Unknown. Data obtained from the electronic medical record included all questions answered during nursing intake and the results of the ID Results form. The nursing intake form asked several yes/no questions including but not limited to, substance use and psychiatric illness history; history of chronic diseases including viral hepatitis and HIV; and infectious diseases screening. Data regarding restrictive housing was also obtained from the electronic medical record. Restrictive housing is a term used to describe when persons are confined to a single-person cell in a separate housing unit and/or the health services unit for administrative, behavioral, or medical reasons.

All elements of this study were approved by the Tufts Health Sciences Institutional Review Board. The approval number for this study is 13059. We reviewed retrospective medical records, and all data were anonymized prior to any research activity. The same institutional review board waived the requirement for informed consent because all data were fully anonymized before the research team accessed them.

## Statistical analysis

There were two outcomes of interest: (1) Requested infectious diseases testing at intake and (2) completion of infectious diseases testing. There was no system to track if the person was given the test result, so this was not an outcome. Testing request at intake (Y/N) was captured on the nursing intake form and completion of testing (Y/N) was determined by results recorded on the ID Results form. Since some people were incarcerated multiple times within the study period, we only analyzed whether a person agreed to or completed testing during their most recent incarceration. Race/ethnicity was evaluated in five main categories: white non-Hispanic, Black non-Hispanic, Hispanic, Asian, and American Indian/Pacific Islander/Other. We recognize that race and ethnicity are two separate entities which should be treated in different categories. However, the data set we accessed grouped race and ethnicity into the five categories. For simplicity, we will refer to these five categories as being race/ethnicity.

For the first outcome, we analyzed data from all persons incarcerated from 2016–2020 (Cohort 1). For the second outcome, we analyzed test result data from 2019–2020 (Cohort 2), because the testing offer changed in 2019 with the start of combined testing for HIV, HCV, and syphilis. People who did not request testing at intake, but completed testing likely requested testing at another time during their incarceration and were directed to the same DPH-community health representative for testing. For both outcomes, we performed univariate and multivariable logistic regression analyses, incorporating variables of interest: age in years, race/ethnicity in the aforementioned groups, place of birth dichotomized into US-born/foreign-born, history of being placed in restrictive housing dichotomized as yes/no, and psychiatric illness history dichotomized as yes/no. We included restrictive housing and psychiatric illness in the model as we hypothesized based on previous work that these factors would represent barriers to testing [37–39]. Multivariable regression models for each outcome were designed using variables that were statistically significant in univariate regression analyses ($p < .05$). If any data were missing, the person was excluded from the analysis, except for the

**Table 1. Demographic characteristics and testing rates for the 2 cohorts of people in jail.**

| | Cohort 1 | Cohort 2 |
|---|---|---|
| | **2016–2020** | **2019–2020** |
| | **N = 8,467** | **N = 3,604** |
| **Age, Mean (SD)** | 39 (11.5) | 38 (11.6) |
| **Days Incarcerated, Mean (SD)** | 43 (100) | 90 (123) |
| **Days Incarcerated, Median (Range)** | 43 (0–1589) | 46 (0–725) |
| **Incarcerated more than once** | 3,824 (45) | 1,249 (35) |
| **Race/ethnicity(%)[1]** | | |
| white Non-Hispanic | 4,856 (57) | 1,970 (55) |
| Black Non-Hispanic | 1,487 (18) | 667 (19) |
| Hispanic | 1,791 (21) | 786 (22) |
| Asian/Pacific Islander | 145 (2) | 75 (2) |
| American Indian/Other | 94 (1) | 94 (3) |
| Missing | 94 (1) | 12 (<1) |
| **Foreign Born** | | |
| No | 6,993 (83) | 2,964 (82) |
| Yes | 1,427 (17) | 607 (17) |
| Missing | 47 (<1) | 33 (1) |
| **Restrictive Housing (%)[2]** | 1,461 (17) | 664 (18) |
| **Self-Identified Psychiatric Illness (%)** | | |
| **No** | 3,671 (43) | 1,419 (39) |
| **Yes** | 4,408 (52) | 1,834 (51) |
| **Missing** | 388 (5) | 351 (10) |
| **History of HIV** | 81 (1) | 36 (1) |
| **History of HCV** | 52 (<1) | 24 (1) |
| **Offered testing for HIV/HCV/Syphilis at intake** | 7,362 (87) | 3,242 (90) |
| | **N = 7.362** | **N = 3,242** |
| **Requested testing at intake** | 1,688 (23) | 812 (25) |
| | **N = 1,688** | **N = 812** |
| **People who completed testing who requested testing at intake** | 271 (16) | 247 (30) |
| | **N = 5,674** | **N = 2,430** |
| **People who completed testing who did not request testing at intake at intake** | 98 (2) | 89 (4) |
| **Total tested** | 369 (4) | 336 (9) |

[1] Race/ethnicity categories were preset in the database from which these data were obtained and individuals were already sorted into these categories. Therefore, we were not able to subdivide individuals into Black Hispanic and white Hispanic.

[2] Restrictive Housing is a term used to describe when persons are confined to a single-person cell in a separate housing unit and/or the health services unit for administrative, behavioral, or medical reasons.

demographic information in Table 1. The data was analyzed using STATA SE 14 (StataCorp, College Station, TX).

# Results

There were 8,467 people incarcerated between 2016–2020 with electronic medical records. These people comprised Cohort 1. Cohort 2 (2019–2020) included 3,604 people. People were not excluded from the data set in Table 1 if they had missing information for any of our variables of interest. There was no significant difference between our outcomes of interest and the

missing and non-missing values of the variables with missing data. This finding indicates that our data were missing at random, and we were able to exclude individuals with missing information from our univariate and logistic regressions. Thus, the sizes of Cohorts 1 and 2 are different in Tables 2 and 3. The demographics for both samples were comparable. The average age in Cohort 1 was 39 years (SD 11.5); The average age of Cohort 2 was 38 years (SD 11.7). White non-Hispanic people made up more than half the sample size, nearly one-fifth had experienced restrictive housing, and more than half reported having a history of psychiatric illness (Table 1).

### Infectious diseases testing request

After exclusions for missing race/ethnicity data, region of birth data, and psychiatric data, we were left with 6,873 people. In Cohort 1 (2016–2020), 23% (1,688) requested testing. In the univariate analysis, increasing age was significantly negatively associated with request of infectious diseases testing (P <0.001) (Table 2). Also in the univariate analysis, Black non-Hispanic and Hispanic race/ethnicity, and ever experiencing restrictive housing were significantly

**Table 2. Univariate and multivariable regression analysis for those who requested testing by characteristics (Cohort 1).**

| | Requested testing at intake | | | |
| --- | --- | --- | --- | --- |
| | No (N = 5,312) | Yes (N = 1,561) | OR (CI) | P-value |
| **Age, mean (SD)** | 40 (12) | 37 (11) | **0.97 (0.97–0.98)** | **<0.001** |
| **Incarcerated more than once (%)** | | | | 0.64 |
| No | 3,218 (77) | 956 (23) | Ref | |
| Yes | 2,094 (78) | 605 (22) | 0.97 | |
| **Race/ethnicity (%)**[1] | | | | **<0.001** |
| white Non-Hispanic | 3,237 (81) | 772 (19) | Ref | |
| Black Non-Hispanic | 908 (75) | 308 (25) | **1.42 (1.22–1.65)** | |
| Hispanic | 1,012 (70) | 439 (30) | **1.82 (1.59–2.09)** | |
| Asian | 89 (75) | 29 (25) | 1.37 (0.89–2.09) | |
| American Indian/Pacific Islander/Other | 66 (84) | 13 (16) | | |
| **Foreign Born (%)** | | | Ref | **0.02** |
| No | 4,473 (78) | 1,271 (22) | **1.20 (1.21–1.60)** | |
| Yes | 839 (72) | 290 (25) | | |
| **Experienced Restrictive Housing (%)**[2] | | | Ref | **<0.001** |
| No | 4,473 (78) | 1,238 (22) | Ref | |
| Yes | 839 (72) | 323 (28) | **1.30 (1.21–1.60)** | |
| **Self-identified Psychiatric Illness (%)** | | | | 0.10 |
| No | 2,362 (76) | 731 (24) | Ref | |
| Yes | 2,950 (78) | 830 (22) | 0.91 (0.81–1.02) | |
| **History of HIV (%)** | | | | 0.10 |
| No | 5,257 (77) | 1,553 (23) | Ref | |
| Yes | 55 (87) | 8 (13) | 0.49 | |
| **History of HCV (%)** | | | | 0.20 |
| No | 5,276 (77) | 1,555 (23) | Ref | |
| Yes | 36 (86) | 6 (14) | 0.57 | |

[1] Race/ethnicity categories were preset in the database from which these data were obtained and individuals were already sorted into these categories. Therefore, we were not able to subdivide individuals into Black Hispanic and white Hispanic.

[2] Restrictive Housing is a term used to describe when persons are confined to a single-person cell in a separate housing unit and/or the health services unit for administrative, behavioral, or medical reasons.

positively associated (P = 0.001) with requesting testing at intake. Age, race/ethnicity, and experience of restrictive housing were included in the multivariable model. The absolute percentages for agreeing to infectious disease testing by race/ethnicity were 30% for Hispanic people, 25% for Black non-Hispanic people, 25% for Asian people, 19% for white non-Hispanic people, and 16% for American Indian/Other race/ethnicity people. In the multivariable model, Black non-Hispanic and Hispanic people had an increase in adjusted odds for requesting testing compared to their white counterparts; 32% (AOR, 1.32; 95% CI, 1.13–1.55) and 69% (AOR, 1.69; 95% CI 1.44–1.99), respectively. Twenty-eight percent of people who had experienced restrictive housing agreed to testing (323) versus 22% who had not experienced restrictive housing (1,238). This translates into 24% increased odds of requesting testing for someone who had experienced restrictive housing (AOR, 1.24; 95% CI, 1.07–1.44). The complete results related to requesting testing of infectious diseases can be found in Table 2.

## Completion of infectious diseases testing

After exclusions for missing race/ethnicity data, region of birth data, and psychiatric data, three-hundred-nine people completed testing during the study period in Cohort 2. Of those who agreed to testing at intake in Cohort 2 (812), 38% received it (309 people). In the univariate analysis, increasing age was significantly negatively associated with completing infectious disease testing (P < 0.001). Also in the univariate analysis, Black non-Hispanic and Hispanic race/ethnicity (P < 0.001) and having experienced restrictive housing (P < 0.001) were significantly positively associated with completing infectious disease testing. Age, race/ethnicity, and experience of restrictive housing were included in the multivariable model. The absolute percentages for completing testing by race/ethnicity were 16% for Hispanic people who were incarcerated, 14% for Black non-Hispanic people, 10% for Asian/Pacific Islander people, 8% for white non-Hispanic, and 6% for American Indian/other race/ethnicity people. In the multivariable model, people who were Hispanic had an increased odds of 69% for having completed infectious diseases testing compared to their white non-Hispanic counterparts (AOR, 1.69; 95% CI, 1.24–2.29). In the multivariable model, people who were Black had an increased odds of 156% for having completed infectious diseases testing compared to their white non-Hispanic counterparts (AOR, 2.56; 95% CI, 1.70–2.99). There were no statistically significant differences found between race/ethnicity groups of Asian/Pacific Islander or American Indian/other with reference race/ethnicity group white non-Hispanic. Twenty-two percent of people who had experienced restrictive housing completed testing, compared to 8% who had not experienced restrictive housing. Those who were housed in restrictive housing had a 217% increase in odds for having completed testing, compared to all other people (AOR, 3.17; 95% CI, 2.46–4.07). All the above results related to completion of infectious disease testing can be found in Table 3.

## Discussion

In the current study, we found that less than a quarter of people entering jail requested infectious diseases testing and only 38% of people who requested infectious diseases testing at intake had testing completed. We found that Black non-Hispanic and Hispanic people detained at a single county jail in Massachusetts were both more likely to request testing for infectious diseases and more likely to complete infectious diseases testing compared to their white non-Hispanic counterparts. Our results show that there is still much work to be done towards the goal of getting people who are incarcerated and detained tested for infectious diseases, a crucial step in HIV and HCV elimination efforts.

**Table 3. Univariate and multivariate regression analysis of those who completed testing by characteristics (Cohort 2).**

| | Received Results | | | | | |
|---|---|---|---|---|---|---|
| | No (N = 2,562) | Yes (N = 309) | OR (CI) | P-value | AOR (CI) | P-value |
| **Age, mean (SD)** | 39 (12) | 35 (11) | **0.97 (0.96–0.98)** | **<0.001** | **0.98 (0.97–0.99)** | **<0.001** |
| **Days incarcerated (%)** | 74 (111) | 198 (150) | **1.01 (1.00–1.01)** | **<0.001** | **1.01 (1.00–1.01)** | **<0.001** |
| **Race/ethnicity (%)**[1] | | | | **<0.001** | | **0.004** |
| white Non-Hispanic | 1,482 (92) | 127 (8) | Ref | | Ref | |
| Black Non-Hispanic | 441 (86) | 72 (14) | **1.91 (1.40–259)** | | **1.54 (1.10–2.15)** | |
| Hispanic | 512 (84) | 99 (16) | **2.56 (1.70–2.99)** | | **1.69 (1.24–2.29)** | |
| Asian/Pacific Islander | 54 (90) | 6 (10) | 1.30 (0.55–3.07) | | 0.88 (0.35–2.20) | |
| American Indian/Other | 73 (94) | 5 (6) | 0.80 (0.32–2.01) | | 0.78 (0.29–2.05) | |
| **Foreign Born (%)** | | | | 0.11 | | |
| No | 2,141 (90) | 247 (10) | Ref | | | |
| Yes | 421 (87) | 62 (13) | 1.28 (0.98–1.72) | | | |
| **Ever experienced restrictive housing (%)**[2] | | | | **<0.001** | | **<0.001** |
| No | 2,134 (92) | 189 (8) | Ref | | Ref | |
| Yes | 428 (78) | 120 (22) | **3.17 (2.46–4.07)** | | **1.66 (1.25–2.21)** | |
| **Self-identified Psychiatric Illness (%)** | | | | 0.05 | | |
| No | 1,085 (88) | 149 (12) | Ref | | | |
| Yes | 1,477 (90) | 160 (10) | 0.79 (0.62–1.00) | | | |
| **History of HIV (%)** | | | | 0.50 | | |
| No | 2,535 (89) | 307 (11) | Ref | | | |
| Yes | 27 (93) | 2 (7) | 0.61 (0.14–2.58) | | | |
| **History of HCV (%)** | | | | 0.85 | | |
| No | 2,543 (89) | 307 (11) | Ref | | | |
| Yes | 19 (90) | 2 (10) | 0.87 (0.20–3.76) | | | |

[1] Race/ethnicity categories were preset in the database from which these data were obtained and individuals were already sorted into these categories. Therefore, we were not able to subdivide individuals into Black Hispanic and white Hispanic.

[2] Restrictive Housing is a term used to describe when persons are confined to a single-person cell in a separate housing unit and/or the health services unit for administrative, behavioral, or medical reasons.

Notably, our findings stand in stark contrast to our hypothesis. Based on a review of the literature, we hypothesized that minoritized communities incarcerated in jail would be less likely to request and complete infectious diseases testing, however our data show that Black non-Hispanic and Hispanic people were more likely to request and complete testing. Black non-Hispanic and Hispanic people who are incarcerated may experience increased barriers to testing in the community compared to white people, so they may be more inclined to accept testing in a carceral setting. One possibility we considered was differential access to health insurance in the community experienced by Black non-Hispanic and Hispanic people. Upon incarceration, people lose their health insurance, and the jail assumes financial responsibility for all their healthcare needs. Most people in Massachusetts theoretically have health insurance due to MassHealth, a public health insurance program available to any Massachusetts resident of the state. Racial differences in health insurance coverage have been noted, with white residents having higher rates of continuous health insurance coverage compared to Black and Hispanic Massachusetts residents [40]. Insurance coverage does not guarantee access, especially given the primary care workforce shortage [41]. Minoritized populations may face increased barriers to accessing preventative and/or primary care in the community, so they may be more likely to utilize these services while incarcerated [42–45].

People who reported a history of psychiatric illness and people who experienced restrictive housing were also more likely to request testing as well as complete testing. These results also run counter to our hypothesis, as we originally thought that psychiatric illness and restrictive housing may present barriers to testing. The psychiatric illness variable was binary and asked at intake. It is possible that people who said they had history of psychiatric illness at intake had increased interactions with healthcare providers on the outside and increased opportunities to learn about the importance of infectious diseases testing. Another possibility is that these individuals perceived themselves at increased risk because of behaviors related to sub-optimally managed psychiatric illness in the community. Similarly, people with psychiatric illness are potentially more likely to experience restrictive housing, which could explain the increase requests. Increased completion of tests for people with psychiatric illness and restrictive housing could be a result of increased interactions with healthcare staff at the jail leading to more opportunities to get testing.

Testing is currently offered in an "opt-in" method. One potential strategy that could be used by all jails to improve testing is offering opt-out testing, with a question at intake like, "We will conduct infectious diseases testing unless you do not want it." The opt-out method of HIV and HCV testing has been endorsed by the CDC due to evidence of increased uptake in minoritized communities [46, 47]. Opt-out testing is a cost-effective intervention shown to increase testing acceptance by 68% and decrease health disparities [17]. Previous research, including randomized controlled trials, has demonstrated the effectiveness of opt-out testing in increasing frequency of testing, but no research has examined the impact of opt-out testing in decreasing disparities in access to testing in incarcerated populations. Professional medical organizations and the Federal Bureau of Prisons support opt-out HCV screening for all people who are incarcerated [2, 18]. Data from testing in emergency rooms shows increased uptake of opt-out HIV testing by Black people compared to white people [48].

Although we were pleased that the racial disparities in requesting and completing infectious disease testing in the community do not appear to exist in the carceral setting where this study took place, it is still concerning that such a low percentage (38%) of people who requested testing at intake had the test completed (Table 1). A smaller percentage (4%) received it later in their incarceration (Table 1). We propose several explanations for this. First, as stated in the methods section, testing is offered in an opt-in fashion during nursing intake at MHOC. The actual testing is completed in a separate meeting with a community health group. The community health group goes to MHOC one to two times per week. So, there was some wait time between agreeing to testing and meeting with the testing providers. During this time, a person could be decarcerated or change their mind about testing. Additionally, the person could be unavailable to attend the appointment with the community health group and be lost to follow-up. In the meeting with the community health group, there is a formal consent process, during which a person might also change their mind and then decline the test. We have no data on how often any of the above outcomes may have occurred, but they likely made it so there were fewer people who actually consented to complete testing with the community health group compared to those who requested testing at intake. Finally, the outcome that we measured was how many ID Result forms were entered into the EMR. There is an established process for collecting blood specimens and completing the ID Results form in the EMR (outlined in the Methods section). There is the potential for error at each hand-off point in this process, whether it be when the specimen is sent to MassDPH, when the results from MassDPH are given to MHOC, or when the results from MassDPH need to be recorded in the EMR by MHOC nurses. Errors at any point in this chain could result in an underestimation of the true number of infectious disease tests completed. All the reasons above likely contributed to the low number of completed tests. More research is needed to determine which of these factors

contributed the most. However, we believe that the multi-step approach with various players (nurses performing intake, community health group employees, MassDPH, nurses entering information into the EMR) created the potential for error and loss of follow-up.

There are two limitations to this study that deserve discussion. First, due to the classification system in the databases available, race and ethnicity have been grouped together in the database, so differential analysis of people who are Black Hispanic and white Hispanic was not possible. Second, information on psychiatric illness data was limited to the data reported during intake, rather than including the entire period of incarceration.

Despite these limitations, our work used a novel dataset to query important and unanswered questions about the frequency of requesting infectious diseases testing at intake and receiving infectious diseases testing in jails. Our work highlights the need for improved protocols in both prioritizing tests through optimizing the methods around offering testing and further examination of the barriers preventing timely access to infectious diseases testing in jails, with the goal of providing universal infectious diseases testing in jails and prisons. We believe there should be a qualitative assessment of the barriers and facilitators to completing infectious diseases testing. This evaluation should include various perspectives on the acceptability of testing, including people who are incarcerated, key stakeholders, carceral nurses, contracted employees who collect consent and specimens, and laboratory staff.

## Supporting information

**S1 File.**
(CSV)

**S2 File.**
(CSV)

## Author Contributions

**Conceptualization:** Alysse G. Wurcel, Sabrina A. Assoumou.

**Data curation:** Alysse G. Wurcel, Rubeen Guardado, Peter J. Koutoujian, Kashif Siddiqi, Thomas Senst.

**Formal analysis:** Alysse G. Wurcel, Rubeen Guardado.

**Funding acquisition:** Alysse G. Wurcel.

**Investigation:** Alysse G. Wurcel.

**Methodology:** Alysse G. Wurcel.

**Resources:** Alysse G. Wurcel.

**Supervision:** Alysse G. Wurcel.

**Writing – original draft:** Alysse G. Wurcel, Rubeen Guardado.

**Writing – review & editing:** Alysse G. Wurcel, Rubeen Guardado, Emily D. Grussing, Peter J. Koutoujian, Kashif Siddiqi, Thomas Senst, Sabrina A. Assoumou, Karen M. Freund, Curt G. Beckwith.

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
