## [Decision Letter · Decision Letter 0]

9 Mar 2023

PONE-D-22-31064Racial Differences in Testing for Infectious Diseases: An Analysis of Jail Intake DataPLOS ONE

Dear Dr. Grussing,

Thank you for submitting your manuscript to PLOS ONE. After careful consideration, we feel that it has merit but does not fully meet PLOS ONE’s publication criteria as it currently stands. Therefore, we invite you to submit a revised version of the manuscript that addresses the points raised during the review process. My apologies for any delays in getting this decision to you.

We look forward to receiving your revised manuscript.

Kind regards,

Andrea Knittel

Academic Editor

PLOS ONE

Journal Requirements:

2. Please provide additional information regarding the considerations  made for the prisoners included in this study. For instance, please discuss whether participants were able to opt out of the study and whether individuals who did not participate receive the same treatment offered to participants.

3. For studies reporting research involving human participants, PLOS ONE requires authors to confirm that this specific study was reviewed and approved by an institutional review board (ethics committee) before the study began. Please provide the specific name of the ethics committee/IRB that approved your study, or explain why you did not seek approval in this case.

5. Please amend your manuscript to include your abstract after the title page.

Reviewers' comments:

Reviewer's Responses to Questions

**Comments to the Author**

1. Is the manuscript technically sound, and do the data support the conclusions?

Reviewer #1: Yes

Reviewer #2: Yes

2. Has the statistical analysis been performed appropriately and rigorously? 

Reviewer #1: Yes

Reviewer #2: Yes

3. Have the authors made all data underlying the findings in their manuscript fully available?

Reviewer #1: Yes

Reviewer #2: Yes

4. Is the manuscript presented in an intelligible fashion and written in standard English?

Reviewer #1: Yes

Reviewer #2: Yes

5. Review Comments to the Author

Reviewer #1: This manuscript addresses an important topic and is well written. Please see attached for my detailed comments regarding ways to improve the manuscript.

Reviewer #2: There are three major issues that should be addressed before publication of this paper:

(1) While the rationale for assessing testing by race is clearly presented, there is no rationale for the other ‘variables of interest.’ How is restrictive housing relevant to HIV/HCV testing? How is psychiatric illness? What is the underlying theory or evidence? Also, 'restrictive housing' needs to be explained more. Is this the same thing as 'solitary'?

(2) Regarding the 2nd cohort, only a third of people who requested testing received it. Were there any differences by race? In other words, were some groups less likely to receive testing after requesting it? This could indicate important differential in service provision by race. This is also relevant to the conclusion in line 226. I’m not sure if you can make the statement about lack of racial inequities UNLESS you show there was no differential by race in requests for testing being honored. (3) authors posit in the abstract and discussion that testing may be higher among Black and Hispanic people because they have longer terms of incarceration. However, the preponderance of testing occurred at intake, so it is unclear why this would be the case.

Additional comments:

Abstract

Spell out MHOC the first time

“This study was a retrospective analysis of people at MHOC that evaluated request and receipt of infectious diseases testing.” Confusing sentence. Clarify.

Intro line 46. It’s not the ‘wording’ of how to offer testing, it’s implementation strategy

Methods line 76. Were people informed they would be tested for syphilis as well as HIV and HCV? Were they asked? Did this influence their requests to be tested?

Results

Line 164. State how many people agreed to testing at intake.

Results line 174, 180. Findings are presented regarding who received results. Authors state earlier in the paper that there are no records of receiving test result (line 104). Which is it?

Discussion

Line 186: The statement that 15% of people offered testing completed should be in results. It is also a bit confusing, because earlier authors state that 1/3 of people who requested testing received it. So does this include people who did not request it? What is the denominator?

Line 200: explain what MassHealth is. This part of the paper seems like a long way of saying that everyone has health insurance in the community, but access may differ by race. Is there any evidence of this?

Line 211: why did authors not incorporate length of time incarcerated in their analysis, if this is a hypothesis?

Technical

Numbers that begin a sentence should be spelled out

Tables 2 & 3 are hard to read. Too many horizontal and vertical lines. Reformat to be similar to tables in PLOS publications.

6. PLOS authors have the option to publish the peer review history of their article (what does this mean?). If published, this will include your full peer review and any attached files.

Reviewer #1: **Yes: **Fiona G. Kouyoumdjian

Reviewer #2: No

---

## [Author Response · Author response to Decision Letter 0]

28 Apr 2023

April 27, 2023

Dear Dr. Andrea Knittel,

Thank you for your thoughtful review of our paper, “Racial Differences in Testing for Infectious Diseases: An Analysis of Jail Intake Data.” We provide responses to your comments below.

Journal Requirements: 

Comment 1: When submitting your revision, we need you to address these additional requirements.1. Please ensure that your manuscript meets PLOS ONE's style requirements, including those for file naming. 

Response 1: Thank you for bringing this to our attention. We have reformatted our documents to follow the guidelines on the document you so kindly shared with us.

Comment 2: Please provide additional information regarding the considerations made for the prisoners included in this study. For instance, please discuss whether participants were able to opt out of the study and whether individuals who did not participate receive the same treatment offered to participants.

Response 2: Thank you for this comment. This was a retrospective analysis of data gathered from an offender management system and electronic medical record. Therefore, people did not provide consent to participate, so there was no opt-in or opt-out protocol. All individuals were included in the study and treated similarly. This protocol was approved by the Tufts University Health Sciences Review Board. The board waived the requirement for informed consent, as we met the parameters they set forth to be a low-risk study because we did not collect identifiable information. 

Comment 3: For studies reporting research involving human participants, PLOS ONE requires authors to confirm that this specific study was reviewed and approved by an institutional review board (ethics committee) before the study began. Please provide the specific name of the ethics committee/IRB that approved your study, or explain why you did not seek approval in this case. Once you have amended this/these statement(s) in the Methods section of the manuscript, please add the same text to the “Ethics Statement” field of the submission form (via “Edit Submission”). For additional information about PLOS ONE ethical requirements for human subjects research, please refer to http://journals.plos.org/plosone/s/submission-guidelines#loc-human-subjects-research.

Response 3: Thank you for this comment. In our methods section, please kindly see our indication that this work was approved by the Tufts University Health Sciences Institutional Review Board. We have ensured that the text in our methods section matches that in the ethics statement. 

Comment 4: We note that the grant information you provided in the ‘Funding Information’ and ‘Financial Disclosure’ sections do not match. When you resubmit, please ensure that you provide the correct grant numbers for the awards you received for your study in the ‘Funding Information’ section.

 Response 4: Thank you for bringing this to our attention. We ensured that the study numbers are correct. I had accidentally selected Tupperware as a funder in the prior submission. This was incorrect, so I removed that funder in this submission. 

Comment 5: Please amend your manuscript to include your abstract after the title page.

Response 5: The abstract has been added (please see lines 65-83).

Reviewer #1:

Comment 1: While it’s interesting to look at restrictive housing and psychiatric illness, you haven’t addressed that content in the Discussion. I suggest you consider cutting that info or else explicitly interpreting that information and discussing the implications.

Response 1: Thank you for pointing out this discrepancy. We have added a justification for including this information in lines 330-332. We then discuss these results in the discussion (please see lines 575-588).

Comment 2: Abstract: I find this sentence unclear: “We hope our collaborative efforts with jail professionals can encourage other cross-disciplinary investigations.”

Response 2: Thank you for this comment. We apologize for any confusion. What we are trying to say in that sentence is that we hope our partnership (one between an academic medical center and jail) can be a model for other researchers to incorporate key stakeholders into their study teams. Oftentimes, in carceral research, people from outside the carceral system come in, do their research, and then publish work without forming long-lasting partnerships or offering author credit to the jail employees. We are trying to model a different professional partnership, as three of our authors on this work are full-time carceral professionals.

Comment 3: Introduction: paragraph 2 (and throughout)- I’m interested to see the use of the term “minoritized” rather than “racialized”- given the explicit focus on race, why not use “racialized”?

Response 3: Thank you for bringing up this interesting point. We have two reasons for using minoritized. First, although you are correct in saying that our main outcome of interest in this work is race, we are also including history of restrictive housing and psychiatric illness in our analysis. So, we wanted a broader term to encapsulate all our outcome variables. Second, we want to highlight to the reader that our outcome variables are directly associated with putting someone in a disadvantaged position. Although racialized may be interpreted similarly by some readers, we were not sure that all readers would come to their own conclusion that having something be racialized inherently reflects systemic prejudice and disenfranchisement. Therefore, we prefer the term minoritized. Please see Husain et al. 2023 to see another work that used minoritized in a similar manner (https://www.sciencedirect.com/science/article/abs/pii/S0740547222002008).

Comment 4: paragraph 2- I think the content in this paragraph from “Disparities in access to HIV and HCV testing…” onward seems unrelated to the content in the first half of the paragraph, including the initial sentence. Consider splitting up the paragraph or revising the initial sentence to improve coherence/flow.

Response 4: Thank you for this comment. We have split this section of text into 2 paragraphs to improve coherence/flow (please see lines 101-131). 

Comment 5: Methods: -line 73- is the language used to ask people if they want testing actually “HCV” or “hepatitis C”? I assume not everyone would know the acronym HCV and so wonder how accessible this question would be.

Response 5: Thank you for pointing out this important topic. We have revised the manuscript to show that people were asked about hepatitis C. Please see line 143.

Comment 6: -line 74- is the “contracted community health group” the MassDPH? I find this unclear. May need to clarify consent+specimen collection vs. testing…; line 79- would be nice to know who does case management and contact tracing, partly to show the value of the jail-based testing

Response 6: Thank you for asking this question. We prefer not to name the group. We did add clarifications about the role of the group and that the group also manages linkage-to-care: “The MassDPH funds a community-based public health agency to come into MHOC weekly and perform testing” and “The DPH also funds linkage-to-care specialists who help link people with HIV and HCV to care in the community after release who come to the jail 1-2 times a week and also manage linkage to HCV and HIV care in the community.” Please see lines 136-191 for these changes.

Comment 7: lines 79-83- I understand that there are multiple opportunities to request testing, but can you clarify whether all testing is conducted by MassDPH? i.e. are the data presented comprehensive for all those tested?

Response 7: Thank you for asking these questions. The jail clinicians have been instructed to perform HIV and hepatitis C screening through the MassDPH. It is possible that occasionally samples are run through the regular laboratory process, but this is unlikely. 

Comment 8: lines 90-91- Should describe whether data on race/ethnicity are self-reported vs. assigned, and who collects these data (e.g. a correctional officer?).

Response 8: Thank you for this question. We have included this information on lines 182-183.

Comment 9: 93-96-why only look at intake data on psychiatric history, recognizing that some people will be diagnosed with or initially report psychiatric illness subsequent to admission? Good to clarify and/or report as a limitation.

Response 9: Thank you for bringing up this question. We have listed this as a limitation (please see lines 604-606).

Comment 10: -90-90 vs. 107-108- why are the cats different in these two sections? Is this the difference between the primary data collection and the dataset?

Response 10: We believe this question is about race. Thank you for pointing out these discrepancies. The correct categories are white, NH; Black, NH ; Hispanic; Asian/Pacific Islander; American Indian/Other. Some individuals were identified as “Asian/Pacific Islander” at the time of the intake, and we could not distinguish whether it was one, the other or both. Then some other individuals who were identified as only “Asian.” Therefore, we grouped these two groups together. We edited the manuscript to be consistent throughout.

Comment 11: lines 116-118- “People who did not request testing at intake, but completed testing, were classified as having completed testing because they likely requested testing at another time during their incarceration.” Should this say “were classified as having requested testing”?

Response 11: Thank you for this question. It should actually be, “People who did not request testing at intake, but completed testing likely requested testing at another time during their incarceration and were directed to the same DPH-community health representative for testing,” (please see lines 237-239).

Comment 12: Results-lines 125-127- Excluding those with any missing data may bias the findings. Why not include all persons and report missing data whenever possible? Especially since this seems to be a fairly large number of people with any missing data (>400 in each cohort) and missingness is likely non-random.

Response 12: Thank you for bringing up this important topic. We edited Table 1 to include missingness. Missingness was reported, but then was assessed for randomness to determine whether or not to include it in the logistic regressions. To test for randomness of missing data, we assessed the relationships between our outcomes of interest and the missing and non-missing values of the variables with missing data using a chi-squared test at an alpha of 0.05. We found there was no significant difference between our outcomes of interest and the missing and non-missing values of the variables with missing data. This indicates that our data is missing at random and therefore was excluded from our univariate and logistic regressions.

Comment 13: line 156-I would provide absolute numbers for those with and without psychiatric illness, in addition to AOR data. Also should use the same language throughout, i.e. is it “mental health history,” “psychiatric illness history,” or “treated psychiatric illness history”? Same issues in Tables…

Response 13: Thank you for pointing this out. We changed the language to be “psychiatric illness history” throughout the manuscript and tables. We have added in the absolute numbers for experiencing restrictive housing. We did not include psychiatric illness results because it was not significant but directed readers to table 2 to find this information.

Comment 14: line 161-162- Aren’t you looking at receipt of testing for all persons and not just those who opted in to testing? i.e. are you looking at receipt of testing conditional on opting in? This isn’t what you describe in the Methods…I think both are important but need to clarify which you are assessing and line up content in the Methods+Results accordingly.

Response 14: Thank you for this comment. You are correct. We were looking at how many people receive testing. We have clarified the manuscript to just report the raw number of people who received testing (please see line 300).

Comment 15: lines 164-165- Could this be an issue of tests vs. people tested? i.e. 449/479. Also, I don’t think you talked about accessing data on tests performed from DPH in Methods- can you reconcile/clarify this?

Response 15: Thank you for pointing out this possibility. We no longer discuss this issue because we changed our unit of measure from “ever tested” to “tested during most recent incarceration.” We have clarified the manuscript to state that the data we accessed was not directly from the DPH, but rather documentation of the number of tests the DPH completed for Middlesex House of Correction.

Comment 16: line 170- You mentioned the conflation of race/ethnicity in the categories, and yet you are using the word race in the text. Why not consistently use race/ethnicity throughout?

Response 16: Thank you for this comment. We have made this change throughout the manuscript and tables. 

Comment 17: line 174, line 180- I think the language re: the second outcome is unclear here- sounds like you are looking at receiving the test result vs. being tested, which is different from the definition of second outcome in line 104. Related to this, consider whether helpful to revise the language throughout to say “Obtained testing” or “Completed testing” rather than “Received testing.” Same issue in Table 3 (“received results”).

Response 17: Thank you for bringing up this point of confusion. Our aim was to report findings for people who have tested completed, not just received the test result. We changed our language to discuss completion of testing rather than receipt of testing to try to clarify this point. 

Discussion 

Comment 18: line 183-185- compared with…

Response 18: Thank you for highlighting this error. We changed this sentence to reflect that the comparison group was white non-Hispanic (please see line 322).

Comment 19: line 186- I wonder if it would be more appropriate to define the first outcome as “opting in to testing” vs. “requesting testing.” Requesting suggests greater initiative on the part of the person in custody than what is actually happening. Also may be good to clarify whether those who requested testing at a time other than intake are included in cohort 1- this isn’t clear to me. (Line 163 says “The remaining completed tests (111) were done at a time outside of intake.” but I’m not sure if this is about the timing of testing vs. opting in/requesting). 

Response 19: Thank you for this suggestion and question. We originally tried to use the term “opt-ed” in, but it did not flow well. We changed the term to “requested testing at intake” for the first outcome, using cohort 1.

Comment 20: lines 188-189- “Our results show that there is still much work to be done towards the goal of getting people who are incarcerated and detained access to infectious diseases testing” Can you clarify what you mean here? I think the data don’t clearly show that access is the issue, vs. acceptability…

Response 20: Thank you for this important comment. What have reworded this sentence to read: “Our results show that there is still much work to be done towards the goal of getting people who are incarcerated and detained tested for infectious diseases, a crucial step in HIV and HCV elimination efforts,” (please see lines 469-472). We certainly agree that acceptability is a factor. 

Comment 21: lines 193-204-Can you be explicit about what you think the goal is for testing? Is it universal testing in custody? testing where there is an unmet need for testing? Also recommend you note that it is a limitation that you don’t have access to data on community-based testing (to look at recent testing) or data on testing indication (e.g. IDU, sexual behaviours, etc.).

Response 21: Yes, we believe universal testing is the goal, as it is the recommended standard of care set forth by the CDC (please see introduction). We did not feel that stating this goal fit well in the section of text highlighted here, but we added this information to the conclusion. We respectfully disagree with this last recommendation for analyzing testing indication. The CDC recommends that all people who are incarcerated be offered this testing because being incarcerated is, in and of itself, an indication for testing. So, we do not believe this is a limitation of our study. In terms of access to community records, we also believe this is not necessarily a limitation, as offering testing is recommended in carceral facilities regardless of last date of testing. 

Comment 22: lines 205-212-You have data on dates of incarceration- couldn’t you do some simple stratified analyses of whether/how length of time in custody is associated with testing as a supplementary analysis? Seems odd to me to speculate about this when you have the data to easily look at it. 

 Response 22: Thank you for asking this question. We have added days incarcerated as a mean and range to Table 1. We have also added days incarcerated into Table 3. 

Comment 23: line 218- acceptability or acceptance?

Response 23: Thank you for bringing this to our attention. We agree, it should be acceptance. We modified the text (please see line 542).

Comment 24: line 248-249- Wouldn’t some of these errors lead to testing not being conducted, e.g. an error in specimens not being sent to MassDPH?

Response 24: We agree with you. We have rephrased this sentence to read “true number of infectious diseases test completed,” to reflect the possibility that a test was not conducted (please see lines 585-588).

Comment 25: lines 258-259- I don’t see why you can’t look at who had HIV and HCV since you said you had access to those data in the Methods. Then you could remove those people from the analysis and focus on those eligible for screening.

 Response 25: Thank you for bringing up this point. We have added history of HIV and HCV diagnoses into tables 1, 2, and 3. We chose not to remove these people from the analysis, though, because even if someone had HIV does not mean that they would not agree to HCV testing or vice versa. Therefore, we chose to not exclude individuals with history of HIV or HCV. 

Comment 26: lines 267-268- The assessment of “ever tested” isn’t clear in the Methods and I think this is problematic. I think you should consider whether to run analyses for one admission (random, most recent, etc.) rather than by person. Alternatively, you should clarify what you did and why in the Methods. 

 Response 26: Thank you for bringing up this concern as well as ways to rectify it. We modified our analysis to be based on most recent incarceration. We added a description to the methods (please see lines 217-219).

Comment 27: 273-277-Given the substantial limitations of use of available administrative data to understand why people aren’t being tested, I think it makes sense to talk about additional work, e.g. engaging with health care workers (those offering opt in testing, those conducting testing) and people in the jail to understand the issues better, which could be through qualitative research or engagement with these groups to support interpretation. What exactly the “improved protocols” would be is very unclear based on the study findings.

Response 27: Thank you for this thoughtful suggestion. Information on next steps has been added to the conclusion. Please see lines 607-617.

Tables:

Comment 28: Table 1- rephrase “ever asked about Infectious Disease Testing” to “Offered opt in testing for HIV/HCV/syphilis”?

Response 28: Thank you for this suggestion. We modified the title of the Tables.

Comment 29: Table 1- final rows- add total tested

Response 29: Thank you for this suggestion. It has been made. Please see Table 1.

Comment 30: Table 2-define “restrictive housing” below the table

Response 30: Thank you for this suggestion. We added the same key from table 1 to tables 2 and 3 which includes a definition for restrictive housing. 

Comment 31: Tables 2 and 3-restrictive housing and psychiatric illness are not sociodemographic characteristics- revise title

Response 31: Thank you for this comment. We have changed the titles of Tables 2 and 3.

Reviewer #2:

Comment 1: There are three major issues that should be addressed before publication of this paper:(1) While the rationale for assessing testing by race is clearly presented, there is no rationale for the other ‘variables of interest.’ How is restrictive housing relevant to HIV/HCV testing? How is psychiatric illness? What is the underlying theory or evidence? Also, 'restrictive housing' needs to be explained more. Is this the same thing as 'solitary'?

Response 1: The preferred term by our correctional colleagues is “restrictive housing.” We added this to the variables of interest because we hypothesized people who have been in restrictive housing would be less likely to get testing (please see lines 535-548).

Comment 2: Regarding the 2nd cohort, only a third of people who requested testing received it. Were there any differences by race? In other words, were some groups less likely to receive testing after requesting it? This could indicate important differential in service provision by race. This is also relevant to the conclusion in line 226. I’m not sure if you can make the statement about lack of racial inequities UNLESS you show there was no differential by race in requests for testing being honored. 

Response 2: Thank you for this question. The questions you are asking are addressed in Table 3, where we stratify the rates of completion of testing by race/ethnicity. As you can see in Table 3, Black and Hispanic patients were the only group who were statistically more likely to complete infectious diseases testing. In line 226, we were saying we were glad to see that Black and Hispanic people were not less likely to receive testing (which is the case in the community—please see the introduction). 

Comment 3: Authors posit in the abstract and discussion that testing may be higher among Black and Hispanic people because they have longer terms of incarceration. However, the preponderance of testing occurred at intake, so it is unclear why this would be the case.

Response 3: Thank you for this question. We posited this because the process to complete testing can be several weeks long, since the health vendor only came to the jail 1-2 times per week to collect specimens for everyone who agreed to testing (see methods). If someone were to be decarcerated shortly after intake, even if they had agreed to testing, they would not have testing completed. Therefore, if someone is incarcerated for a longer period of time, it is more likely that they will be able to meet with the health vendor and have testing completed. However, we were able to factor length of incarceration into our new analysis, so we no longer posited on this topic.

Abstract

Comment 4: Spell out MHOC the first time

Response 4: Thank you for this comment. This has been modified (please see line 69).

Comment 5: “This study was a retrospective analysis of people at MHOC that evaluated request and receipt of infectious diseases testing.” Confusing sentence. Clarify.

Response 5: Thank you for your comment. This has been modified to be clearer: “This is a retrospective analysis of people incarcerated at MHOC who opted-in to infectious diseases testing between 2016-2020,” (please see lines 70-71).

Comment 6: Intro line 46. It’s not the ‘wording’ of how to offer testing, it’s implementation strategy

Response 6: Thank you for sharing this thought. We removed the word “wording.”

Comment 7: Methods line 76. Were people informed they would be tested for syphilis as well as HIV and HCV? Were they asked? Did this influence their requests to be tested?

Response 7: Thank you for these questions. Yes, people were asked and consented for all three infections. We do not know how this influenced their request to be tested, as this was unfortunately only a quantitative assessment of testing, not a qualitative one. 

Results

Comment 8: Line 164. State how many people agreed to testing at intake.

Response 8: Thank you for this comment. We have added this (please see line 305).

Comment 9: Results line 174, 180. Findings are presented regarding who received results. Authors state earlier in the paper that there are no records of receiving test result (line 104). Which is it?

Response 9: Thank you for this comment. Reviewer #1 also realized this. We have tried to clarify this section of the manuscript to make it clear that we did not collect data on who received test results. In order to make this section, and the rest of the manuscript, easier to understand, we changed our language from “received testing” to “completed testing” so that the reader would not conflate receiving testing with receiving a result.

Discussion

Comment 10: Line 186: The statement that 15% of people offered testing completed should be in results. It is also a bit confusing, because earlier authors state that 1/3 of people who requested testing received it. So does this include people who did not request it? What is the denominator?

Response 10: Thank you for pointing out this error. Since we redid our analyses to be based on most recent incarceration and not “ever tested,” these numbers are now different. We have reported the new found percentage of people who completed testing in the results (please see lines 408-409).

Comment 11: Line 200: explain what MassHealth is. This part of the paper seems like a long way of saying that everyone has health insurance in the community, but access may differ by race. Is there any evidence of this?

Response 11: Thank you for asking these important questions. We have briefly defined MassHealth, and included a source about how white MA residents are more likely to be insured than Black and Hispanic ones (please see lines 514-534).

Comment 12: Line 211: why did authors not incorporate length of time incarcerated in their analysis, if this is a hypothesis?

Response 12: Thank you for bringing this to our attention. We have modified our Tables to include length of incarceration. 

Technical

Comment 13: Numbers that begin a sentence should be spelled out

Response 13: Thank you for this comment. We have amended this throughout the manuscript. 

Comment 14: Tables 2 & 3 are hard to read. Too many horizontal and vertical lines. Reformat to be similar to tables in PLOS publications.

Response 14: Thank you for sharing this thought. We have removed the horizontal lines in these tables.

Thank you again for the opportunity to revise and resubmit this work.

Sincerely,

Emily Grussing

---

## [Decision Letter · Decision Letter 1]

22 Jun 2023

Racial differences in testing for infectious diseases: An analysis of jail intake data

PONE-D-22-31064R1

Dear Dr. Grussing,

We’re pleased to inform you that your manuscript has been judged scientifically suitable for publication and will be formally accepted for publication once it meets all outstanding technical requirements.

Kind regards,

Andrea Knittel

Academic Editor

PLOS ONE

Additional Editor Comments (optional):

Reviewers' comments:

Reviewer's Responses to Questions

**Comments to the Author**

1. If the authors have adequately addressed your comments raised in a previous round of review and you feel that this manuscript is now acceptable for publication, you may indicate that here to bypass the “Comments to the Author” section, enter your conflict of interest statement in the “Confidential to Editor” section, and submit your "Accept" recommendation.

Reviewer #1: All comments have been addressed

2. Is the manuscript technically sound, and do the data support the conclusions?

Reviewer #1: Yes

3. Has the statistical analysis been performed appropriately and rigorously? 

Reviewer #1: Yes

4. Have the authors made all data underlying the findings in their manuscript fully available?

Reviewer #1: No

5. Is the manuscript presented in an intelligible fashion and written in standard English?

Reviewer #1: Yes

6. Review Comments to the Author

Reviewer #1: (No Response)

7. PLOS authors have the option to publish the peer review history of their article (what does this mean?). If published, this will include your full peer review and any attached files.

Reviewer #1: **Yes: **Fiona G. Kouyoumdjian

---

## [Editor Report · Acceptance letter]

7 Aug 2023

PONE-D-22-31064R1 

Racial differences in testing for infectious diseases: An analysis of jail intake data 

Dear Dr. Grussing:

I'm pleased to inform you that your manuscript has been deemed suitable for publication in PLOS ONE. Congratulations! Your manuscript is now with our production department. 

Kind regards, 

on behalf of

Dr. Andrea Knittel 

Academic Editor

PLOS ONE